# Phylosymbiosis: The Eco-Evolutionary Pattern of Insect–Symbiont Interactions

**DOI:** 10.3390/ijms242115836

**Published:** 2023-10-31

**Authors:** Man Qin, Liyun Jiang, Gexia Qiao, Jing Chen

**Affiliations:** 1Key Laboratory of Zoological Systematics and Evolution, Institute of Zoology, Chinese Academy of Sciences, Beijing 100101, China; qinman@ioz.ac.cn (M.Q.); jiangliyun@ioz.ac.cn (L.J.); 2College of Life Sciences, University of Chinese Academy of Sciences, Beijing 100049, China

**Keywords:** microbial community structure, phylosymbiosis pattern, stochastic effect, codiversification, ecological filtering

## Abstract

Insects harbor diverse assemblages of bacterial and fungal symbionts, which play crucial roles in host life history. Insects and their various symbionts represent a good model for studying host–microbe interactions. Phylosymbiosis is used to describe an eco-evolutionary pattern, providing a new cross-system trend in the research of host-associated microbiota. The phylosymbiosis pattern is characterized by a significant positive correlation between the host phylogeny and microbial community dissimilarities. Although host–symbiont interactions have been demonstrated in many insect groups, our knowledge of the prevalence and mechanisms of phylosymbiosis in insects is still limited. Here, we provide an order-by-order summary of the phylosymbiosis patterns in insects, including Blattodea, Coleoptera, Diptera, Hemiptera, Hymenoptera, and Lepidoptera. Then, we highlight the potential contributions of stochastic effects, evolutionary processes, and ecological filtering in shaping phylosymbiotic microbiota. Phylosymbiosis in insects can arise from a combination of stochastic and deterministic mechanisms, such as the dispersal limitations of microbes, codiversification between symbionts and hosts, and the filtering of phylogenetically conserved host traits (incl., host immune system, diet, and physiological characteristics).

## 1. Introduction of Phylosymbiosis

Host–microbe symbioses play a crucial role in the ecological and evolutionary history of animals [1,2]. Recent advances in the field of host–microbe interactions have demonstrated the influence of host phylogeny and ecological factors on microbial community assembly [3,4,5]. Phylosymbiosis occurs when host-associated microbiota relationships are positively associated with host phylogenetic relatedness.

Phylosymbiosis is defined as “microbial community relationship parallels the host phylogeny”, in which “phylo” refers to host lineage and “symbiosis” refers to the coexistence of hosts and microbes (Figure 1) [6,7]. In other words, microbial community composition dissimilarities are positively associated with the accumulation of host genetic variation. Phylosymbiosis studies focus on the entire microbiota rather than individuals within the microbiota. The persistent and intimate association between microbes and their host is not the necessary assumption of this eco-evolutionary pattern [8].

Pioneering studies on phylosymbiosis were performed on the parasitoid wasp genus *Nasonia* under rearing conditions [9], in which species-specific phylosymbiotic gut bacterial communities caused lethality in interspecific hybrids [7]. Afterward, Brooks et al. [10] revealed phylosymbiosis in other animals, including deer mice (*Peromyscus*), fruit flies (*Drosophila*), and mosquitoes (i.e., *Anopheles*, *Aedes*, and *Culex*). To date, interspecific phylosymbiotic structures of microbiota have been widely reported in insects, birds, fishes, and mammals [5,10,11,12,13,14,15,16,17,18]. However, phylosymbiosis remains poorly understood at the intraspecific level. Intraspecific phylosymbiosis has only been substantiated in the microbial communities from the American pika *Ochotona princeps* [19] and the aphid *Mollitrichosiphum tenuicorpus* [20]. The host taxa in insect phylosymbiosis studies to date cover orders, families, genera, and species, and the evolutionary history of hosts spans approximately 0.3–300 million years [6,21]. The strength of the phylosymbiotic signals between the host and microbiota varies across host taxa [8], and the phylosymbiotic relationships can be weakened with an increasing host evolutionary history [5,21].

Phylosymbiosis analyses typically employ 16S rRNA gene amplicon sequencing data as the input data of the microbial community. Multiple beta diversity distance metrics are usually required for the robustness of the results [8]. Furthermore, a reliable host phylogenetic tree is essential for the determination of phylosymbiosis patterns. The key to measuring phylosymbiosis is to assess the significant correlation between host phylogeny and microbiota beta diversity. Principal methods for quantifying phylosymbiosis are as follows: (1) topological congruency tests [10] utilizing the Robinson–Foulds metric [22] or matching cluster metric [23], or (2) a matrix correlation-based approach, e.g., the Mantel test [24] and Procrustean superimposition [25]. More details on examining phylosymbiosis have been reviewed by Lim and Bordenstein [6].

## 2. Phylosymbiosis in Insects

Insects constitute the most diverse group of animals and play crucial roles in terrestrial ecosystems [26]. Insects harbor a great variety of symbionts, which contribute significantly to the survival, growth, and fecundity of the host [2,27,28]. Additionally, symbionts could facilitate host adaptation to new ecological niches and potentially drive speciation in insects [1,29]. Insect microbial community structures have been found to be correlated with environmental habitat, diet, sex, life stage, and host insect identity and phylogeny [30,31,32]. Some studies highlighted the strongest impact of insect species on the associated microbial communities [30,33]. Currently, phylosymbiosis research in insects remains in its infancy, and phylosymbiosis has been confirmed in the orders Blattodea, Coleoptera, Diptera, Isoptera, Hemiptera, and Hymenoptera (Table 1).

### 2.1. Blattodea

Most insects in the order Blattodea, such as cockroaches and termites, have been characterized by hundreds of species of microbes [48,49,50]. Although omnivorous cockroaches harbor generally similar gut bacterial communities, this ancient lineage and its gut microbiota exhibited a weak but significant phylosymbiotic relationship [21]. In this study, host phylogeny exerted a more important impact on the gut microbiota over a shorter evolutionary history. In termites, several studies have highlighted the variation in intestinal bacteria according to the host phylogeny [51,52,53]. However, the gut bacterial community structures of wood-feeding cockroaches and termites were driven by certain major events in host evolution, such as dietary diversification rather than host phylogeny [54].

### 2.2. Coleoptera

Coleoptera constitutes the most species-rich insect order and depends on various gut bacterial and fungal symbionts to enable plant cell wall digestion [55], plant secondary metabolite detoxification [56], and nutrient provision [57,58]. Research based on beetle species representing five families (i.e., Carabidae, Staphylinidae, Curculionidae, Chrysomelidae, and Scarabaeidae) found that the primary factor shaping the bacterial community is the trophic guild to which the host belongs [59]. Within one specific family, such as dung beetles (Scarabaeinae), host phylogeny and gut morphology had a stronger impact on the gut bacterial community composition than diet [60]. Nevertheless, phylosymbiosis analyses of bacterial communities in coleopterans have yet to be conducted. Regarding symbiotic fungi, phylosymbiotic mycetangial communities were reported in the *Dendroctonus frontalis* species complex (Curculionidae: Scolytinae), but the mechanisms establishing fungal phylosymbiosis were unclear [34].

### 2.3. Diptera

Most dipterans feed on a wide variety of materials, and their symbioses with bacteria have been extensively documented in model systems, including mosquitoes and fruit flies [61,62,63]. The phylosymbiosis pattern has been confirmed in both wild and lab-reared mosquito species [10,36]. *Drosophila* fruit flies exhibit mixed evidence for phylosymbiosis [10,64,65]. Under rearing conditions, host phylogenetic relatedness was positively associated with bacterial community dissimilarities in *Drosophila* flies [10]. However, other studies did not observe phylosymbiosis patterns in wild or laboratory-reared *Drosophila* species [64,65,66]. The variation in gut bacterial communities among *Drosophila* species could be related to responses to different selective pressures rather than host phylogeny, such as geography and diet [67,68]. Within the leaf miner flies of *Liriomyza* (Diptera: Agromiyzidae), the bacterial communities did not exhibit apparent phylosymbiotic signals but were primarily structured by host species identity [69].

### 2.4. Hemiptera

Insects feeding on phloem or xylem sap engage in intimate associations with obligate symbionts, which provide nutritional supplements for the host [29,70]. In addition to obligate symbionts, hemipteran insects possess a variety of facultative symbionts with ecological benefits [71]. These multipartner symbioses have been documented in many hemipteran taxa, including aphids [72,73], psyllids [74,75], whiteflies [76,77], and cicadas [78,79]. Host relatedness was proposed to be an important factor influencing the structures of bacterial communities within Aphidoidea [80]. Phylosymbiosis was observed in the monoecious aphids of Greenideinae at different taxonomic scales, including at subfamily, genus, and intraspecific levels [20,33,37]. However, the dissimilarities in symbiont communities did not correlate significantly with host genetic variation in the heteroecious aphid lineages Eriosomatinae [81] and Hormaphidinae [82]. Another study concerning phylosymbiosis in Hemiptera focused on the psyllid bacterial community, which demonstrated the greater effect of host phylogeny than host plant and geographic distribution [39].

### 2.5. Hymenoptera

Symbioses with bacteria and fungi have been documented in many herbivorous hymenopteran species [83,84,85]. Some gut bacterial symbionts of social bees, such as *Gilliamella apicola*, play key roles in pollen digestion, toxin metabolism, and pathogen protection [86,87]. In fungus-growing ants, cultivated fungi serve as the sole nutritional source for the larvae of attine ants [88]. Phylosymbiosis has been revealed in several Hymenoptera groups, i.e., fig wasps (*Ceratosolen*), parasitoid wasps (*Nasonia*), turtle ants (*Cephalotes*), and *Formica* ants [10,41,42,44]. Under controlled rearing conditions, the bacterial communities of parasitoid wasps of *Nasonia* mirror the host phylogeny at different developmental stages [9]. Within another holometabolous hymenopteran group, the social turtle ants (*Cephalotes*), however, only adults exhibit phylosymbiotic gut bacterial communities; the microbiota of larvae are dominated by environmental bacteria [41].

### 2.6. Lepidoptera

The composition, diversity, and function of gut microbiota in Lepidoptera has been reviewed [89]. The high variability in the composition and diversity of lepidopteran gut microbiota may arise from the environment, host diet, host developmental stage, or host gut physiology [90]. For example, caterpillar larvae acquire a low number of intestinal bacteria and fungi from host plants (e.g., armyworm *Spodoptera frugiperda*) [91] or soil (e.g., cabbage moth *Mamestra brassicae*) [92], while butterflies usually harbor a large number of microbes in their midguts, which are derived from dietary sources [93,94]. Lepidopteran phylosymbiosis has been confirmed in heliconiine butterflies [46]. Hammer et al. [46] suggested that multiple filtering of phylogenetically conserved host traits, including pollen feeding, might have given rise to their phylosymbiotic microbiota.

## 3. Mechanisms Underlying Phylosymbiosis

In most animal systems, microbial transmission and host filtering are major factors influencing microbial community assembly [8,95]. The maintenance of microbes within insect populations usually relies on vertical and horizontal transmission. Strict vertical transmission can promote host–microbe codiversification and ensure the high fidelity of close host–microbe associations during a long evolutionary history. Horizontal transfer of microbes can occur between different individuals of the same or different host species. Horizontal transmissions within conspecifics improve the probability of convergence in microbiota, which may facilitate the appearance of phylosymbiosis [95,96]. However, horizontal transmissions between different host species may weaken the stability of long-lasting host–microbe associations and obscure the phylosymbiotic signatures of microbial communities. For example, significant phylogenetic correlations were not found within the bacterial communities of heteroecious aphids, in which frequent horizontal transmissions of secondary symbionts might have occurred [81,82]. Two typical patterns constitute another principal factor that shapes microbial communities, namely, microorganism filtration within the host. One is the species assortment assembly process, which emphasizes interspecific competition between microorganisms [97]. That is, microbial communities structured according to the species assortment model usually consist of microorganisms that occupy non-overlapping niches. The other is the habitat-filtering model, in which members of the microbiota with similar nutritional requirements tend to arise simultaneously [98,99]. In human gut microbiota with a phylosymbiotic signature, habitat filtering plays a more important role than species assortment [100].

Here, we summarize the contributions of stochastic effects and deterministic forces (i.e., evolutionary and/or ecological factors) on governing the phylosymbiosis patterns in insects (Figure 2). Stochastic and deterministic effects are not mutually exclusive and can contribute to the phylosymbiotic microbiota in combination. For instance, phylosymbiosis in the ants of *Cephalotes* was attributed to a mix of environmental filtering and shared evolutionary history between ants and symbionts [41].

### 3.1. Stochastic Effects

Phylosymbiotic microbiota can be a consequence of stochastic effects, such as spatial limitations on microbial dispersal and random fluctuations in the abundance of microbes (Figure 2A) [101]. Dispersal is referred to as the movement and successful colonization of microbes across space [102]. Moeller et al. [103] revealed that the dispersal limitations of bacteria could promote the compositional divergence of gut microbial communities among mammalian species. In addition to spatial limitations, the composition of the microbial community can be disturbed by the rate and order of microbes that are added to the microbiota during dispersal processes [104]. The microbial dispersal associated with insects generally occurs in the extracellular transmission of microbes, including environmental acquisition, social behavior acquisition, coprophagy, smearing of the egg surface, and capsule or jelly-like secretion transmission [105].

Ecological drift leads to random variation in the relative abundance of species within the microbial community over time [106]. Microbes in low abundances are more susceptible to drift with subsequent extinction. Ecological drift can generate differences in microbial community composition when deterministic processes are weak [104]. In insects, microbiota profiling varies greatly across different groups, with extremes represented by some sap-feeding insects having few gut microbes but abundant intracellular symbionts and by detritivores and wood feeders harboring large and complex gut microbiota [50]. Currently, the effect of ecological drift as the sole factor structuring the microbiota has not been confirmed in any animal system. The phylosymbiotic microbial communities of insects are typically composed of diverse microbes, some of which are abundant and resident. Therefore, the phylosymbiosis pattern within insects is unlikely to be merely drift-driven. Ecological drift may play a part in the interactions with other community assembly processes in structuring insect microbiota.

### 3.2. Evolutionary Processes

Phylosymbiosis can arise from long-term and stable associations between microbes and hosts, such as coevolution and cospeciation. Here, we use “coevolution” in the narrow sense, which emphasizes the reciprocity and simultaneity of evolutionary changes in interacting species [107]. Cospeciation can result from coevolution and occurs when hosts and microbes speciate simultaneously [108]. Demonstrating the coevolution of animals and symbionts under controlled conditions with laboratory models is difficult because it usually requires long periods of time. However, by utilizing phylogenetic and genomic analyses, we can deduce insect–symbiont coevolution [109,110]. Insects feeding on phloem sap, such as species of Hemiptera, possess symbionts that can provide nutrients to compensate for deficiencies in their food source [111,112]. Many hemipteran taxa and their bacterial endosymbionts rely on the biosynthetic and metabolic complementarity of essential nutrition to maintain intimate associations [29,113,114,115]. For instance, the primary endosymbiont *Buchnera aphidicola* has highly coadapted to and evolved with aphids for millions of years [116,117,118]. Likewise, such coevolutionary examples have been identified from extracellular gut symbionts that enable nutrient provisioning, e.g., *Ishikawaella capsulate* in plataspid stinkbugs [119] and *Rosenkranzia clausaccus* in acanthosomatid stinkbugs [120].

Codiversification represents another evolutionary process that underlies phylosymbiosis (Figure 2B). It occurs when hosts and microbes exhibit congruent phylogenetic trees but does not necessarily imply an occurrence of coevolution [121]. Codiversification can be a consequence of unidirectional selection; that is, microbes adapt to the evolutionary changes imposed by their hosts but not vice versa. In the social corbiculate bees, a strain-level phylogenetic association between the core gut bacteria *Lactobacillus* Firm-5 and the host bees was observed, which suggested host–microbe codiversification [122]. Other adaptation processes, such as host-shift speciation [123] and shared geographic isolation [124], can also contribute to matching phylogenies of microbes and host lineages.

Considering the low probability of the entirety of a microbial community being transmitted from mother to offspring with high fidelity, it seems unlikely that all microbiota members are involved in the aforementioned evolutionary processes driving phylosymbiosis. Early-arriving species can affect the ability of late-arriving species to establish themselves during community assembly, which is referred to as priority effects [125]. The importance of priority effects in shaping microbial community composition has been reviewed [126]. Moreover, multiple studies have revealed that highly connected keystone or hub microbes can determine the overall community structure via interspecific interactions [127,128,129]. The evolutionary processes underlying phylosymbiosis represented by coevolution rely on vertical transmissions to maintain the stable inheritance of “early-arriving species”. Heritable symbionts have proven to be universal in herbivorous insects [110,130]. For example, *Buchnera* is located in specialized bacteriocytes and maintained within aphid generations via direct maternal transmission [131]. In the green rice leafhopper *Nephotettix cincticeps*, the facultative symbiont *Rickettsia* is vertically transmitted to offspring paternally via an intrasperm passage as well as maternally via an ovarial passage [132]. Additionally, some extracellular gut symbionts can be maternally transmitted through host generations, such as the specific clade of γ-proteobacteria from acanthosomatid stinkbugs, which is maternally transmitted via egg smearing [133]. For social insects, e.g., *Acromyrmex* leaf-cutting ants [134] and the honey bee *Apis mellifera* [135], social acquisition of beneficial microbes is critical for specificity and partner fidelity in host–bacterial associations. These initial colonizing symbionts with vertical transmission may have served as keystones or hubs and are responsible for the host-species-specific microbial community composition, which provides the opportunity for phylosymbiosis to occur.

### 3.3. Ecological Filtering

Moran and Sloan [121] proposed that phylosymbiosis patterns could emerge from simple ecological filtering without any long-term coevolutionary mechanisms. In principle, some host traits can function as filters that exert a selective role on environmental microbes, and the microbes suitable according to these selective forces can coexist with the host (Figure 2C). It is possible that hosts maintain host-species-specific microbial communities via a strong selection of environmental microbes and then yield phylosymbiotic microbiota. Closely related hosts have similar physiological characteristics, immune systems, or microbial defense mechanisms [133,136,137,138], which may bring about the tendency to harbor similar microbial communities. If the ecological factors that shape microbiota structures are highly phylogenetically conserved during host evolutionary history, we can observe a phylosymbiotic relationship between the host and microbiota [8]. Here, we provide several potential ecological factors shaping the phylosymbiotic microbiota of insects.

#### 3.3.1. Immune System

Numerous studies have highlighted the importance of the host immune system in regulating microbial community composition [139,140,141,142]. Insects rely on physiological barriers and innate immune responses to defend themselves against pathogens [143,144]. The innate immune system of insects is composed of cellular immune responses by circulating hemocytes [145] and humoral immune responses. Although the hemocyte categories involved in the cellular immune responses vary among different insect species, hemocyte functions primarily include phagocytosis, nodulation, and encapsulation [146,147,148]. The humoral defenses are modulated by the Toll, immune deficiency (IMD), Jun N-terminal kinase (JNK), Janus kinase/signal transducers and activators of transcription (JAK/STAT), and prophenoloxidase (PPO) pathways [144,149]. The expression of genes in these pathways subsequently results in antimicrobial peptide (AMP) production, reactive oxygen species (ROS) generation, and melanization. Insects depend on two pathways to regulate antimicrobial peptide generation, namely, the Toll pathway, which responds to fungi and most Gram-positive bacteria, and the IMD pathway, which is induced by Gram-negative bacteria [150].

An insect’s innate immune system not only defends against pathogens but also plays an important role in maintaining host–microbe symbiosis [151,152,153]. Serving as one of the model systems in Hemiptera, aphids lack several immune-related genes that are suspected to be essential in arthropod immunity [154]. Previous studies have suggested that the reduced antimicrobial defense in aphid immunity is attributed to the maintenance of symbionts [155]. To be more specific, the extent of alteration in multiple aphid cellular immunity responses is related to the difference in facultative symbiont species [156]. Eusocial corbiculate bees, including honey bees, bumblebees, and stingless bees, harbor distinctive gut microbiota that are more similar among closely related bee species [122]. The exotic strain of the gut symbiont *Gilliamella* in honey bees induced higher prostaglandin (PG) production than the native strain, which increased the expression of genes in the IMD and Toll immune pathways [157]. These immune pathways then modulated dual oxidase (Duox) production and ROS generation to inhibit the non-native strain of *Gilliamella*.

#### 3.3.2. Diet

There is increasing evidence that diet plays a pivotal role in shaping the microbiota structures of animals [16,32,158,159]. Diet has emerged as a key filter of mammalian gut microbiota [160,161]. The gut microbiota of non-flying mammals was strongly correlated with diet and host phylogeny [159]. Likewise, the microbial communities in bamboo-feeding insects were filtered by diet [31]. If diets themselves are phylogenetically non-independent, they can serve as ecological filters and lead to phylosymbiotic microbiota. Moreover, complete dietary shifts over long evolutionary periods can disrupt the phylosymbiotic relationships between hosts and microbial communities [5].

Host plants are one of the major ecological factors shaping the bacterial communities of insect herbivores [31,32,162,163]. The gut microbial communities of caterpillars are dominated by transient and diet-associated bacteria [164], whereas major members of the adult-stage gut microbiota in butterflies are abundant and consistent [46]. The phylosymbiotic signature of microbiota within heliconiine butterflies may arise from the filtering of phylogenetically conserved diet preferences [46]. Within aphids, host–symbiont codiversificationm as well as filtering by host plants, has been highlighted in structuring the phylosymbiotic microbiota of Greenideinae species [37].

#### 3.3.3. Physiological Characteristics

Another candidate ecological filter underlying host species-specific microbiota is the host’s physiological structure, such as the gut [165] and proventriculus [166]. The biomolecules such as glycans and mucins secreted by the host intestinal wall shape different intestinal environments and are regulators of gut microbial community composition [167,168]. Other host-specific physical and chemical factors in the gut, including the biochemical characteristics of the intestinal surface, pH, oxygen levels, and concentrations of metabolites, are also potential filtering factors of microbes. If these factors themselves are phylogenetically conserved over evolutionary history, the microbial communities might exhibit significant correlations with host phylogeny.

The selective filtering of microbes in the gut environment can explain major variations in phylosymbiotic gut microbial communities in humans [100]. Compared with mammals, birds (e.g., cranes) have strong gastric acidity, which can serve as a microbial filter to limit host-associated differentiation in the gut microbiota and subsequently result in weak phylosymbiotic signatures [14]. In insects, selective effects of the gut environment were experimentally confirmed in the cockroach gut microbiota [169]. Cockroaches preferentially select bacteria that are specifically adapted to their intestinal environment. The proventricular filtering mechanism in ants is responsible for the maintenance of ant–bacteria fidelity [166]. Although the importance of the host’s physiological characteristics in filtering gut microbiota has been emphasized in certain insect groups, its role in shaping insect phylosymbiosis remains poorly understood.

## 4. Future Directions for Research on Phylosymbiosis in Insects

While host–symbiont interactions have been documented across many insect groups, we still have a poor understanding of the prevalence of phylosymbiosis in insects. Phylosymbiotic investigations should be performed on a greater variety of insects to sufficiently disentangle the mechanisms underlying this pattern. In addition to bacterial and fungal communities, phylosymbiosis studies at the insect–virome level [170,171] will contribute to developing a comprehensive landscape of host–microbe symbioses. The application of metagenomic sequencing data to phylosymbiosis detection is recommended due to its finer-scale taxonomic and functional profiling. Integrated multi-omic analyses of the microbiome are advantageous in comprehending the mechanisms behind phylosymbiosis because they resolve linkages between host functions, microbial diversity, microbial functions, and environmental variables [172].

To date, most studies have focused on the impact of evolutionary processes on driving phylosymbiotic microbiota. Quantifying the contribution of ecological filtering factors in phylosymbiosis will greatly improve our understanding of the mechanisms behind these patterns. Host-specific biological characteristics and environmental factors should be identified and evaluated quantitatively in the future. It is more likely that a combination of multiple mechanisms rather than a single evolutionary or ecological process is involved in the development of phylosymbiosis; therefore, candidate mechanisms, including stochastic effects, evolutionary processes, and ecological filtering, need to be tested in a diversity of symbiosis systems in both evolutionary and ecological contexts.

## Figures and Tables

**Figure 1 ijms-24-15836-f001:**
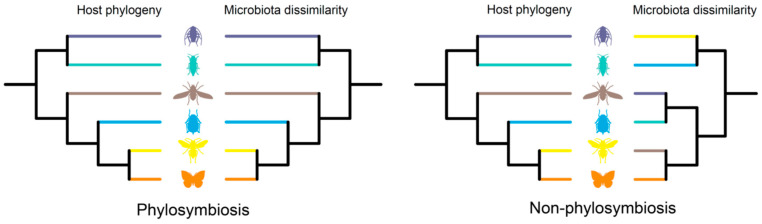
Phylosymbiotic versus stochastic microbial community assemblages. Branches in the same color indicate the host and associated microbial community.

**Figure 2 ijms-24-15836-f002:**
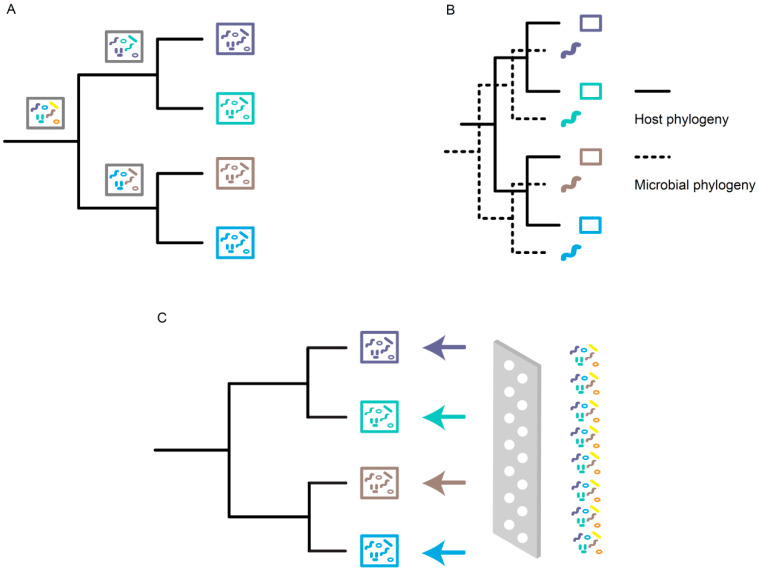
Mechanisms underlying phylosymbiosis. (**A**) Gain or loss of microbes arises from stochastic processes. (**B**) Evolutionary processes, such as codiversification, yield congruent phylogenetic trees of hosts and microbes. (**C**) Ecological filters select suitable environmental microbes to coexist with the hosts.

**Table 1 ijms-24-15836-t001:** Summary of phylosymbiosis patterns in insects.

Insects Examined		No. of Species Sampled	Evolutionary Time (Mya)	Diet	Core Microbe	Obligate Symbiont	References
Blattodea		19	>300	Omnivory	Bacteroidetes, Firmicutes, and Proteobacteria	—	[21]
Coleoptera	*Dendroctonus frontalis* species complex	7	12	Phloem cell	*Ceratocystiopsis*	—	[34,35]
Diptera	*Anopheles*, *Aedes*, and *Culex*	8	100	Blood	Proteobacteria	—	[10,36]
	*Drosophila*	6	63	Decaying fruit	Proteobacteria	—	[10]
Hemiptera	Greenideinae	53	83	Phloem sap	—	*Buchnera aphidicola*	[37,38]
	*Mollitrichosiphum*	8	18–19	Phloem sap	—	*Buchnera aphidicola*	[33]
	*Mollitrichosiphum tenuicorpus*	1 (26 colonies)	11	Phloem sap	—	*Buchnera aphidicola*	[20]
	Psylloidea	102	350	Phloem sap	—	*Carsonella ruddii*	[39,40]
Hymenoptera	*Cephalotes*	13	46	Pollen and honeydew	—	*Cephaloticoccus*	[41]
	*Ceratosolen*	6	60	Fig	*Wolbachia*	—	[42,43]
	*Formica*	14	30	Honeydew and nectar	*Wolbachia*, *Lactobacillus*, *Liliensternia*, and *Spiroplasma*		[44,45]
	*Nasonia*	4	<1	Fly puparium	Proteobacteria, Firmicutes, and Actinobacteria	—	[10]
Lepidoptera	Heliconiini	23	20–30	Pollen, nectar, and fruit	*Acinetobacter*, *Apibacter*, *Asaia*, *Commensalibacter*, *Enterobacter*, *Enterococcus*, *Lactococcus*, *Spiroplasma*, and *Pseudomonas*	—	[46,47]

## Data Availability

Not applicable.

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
