# Peer review of "Phylosymbiosis: The Eco-Evolutionary Pattern of Insect–Symbiont Interactions"

_ijms, 2023, doi:10.3390/ijms242115836_

Round 1

Reviewer 1 Report

Comments and Suggestions for Authors

This publication is a review providing a summary of patterns of phylosymbiosis in insects. The authors put a lot of work into explaining the issue of phylosymbiosis and describing research on this topic. They present examples of phylosymbiosis in various orders of insects and describe the mechanisms underlying phylosymbiosis. This publication is quite interesting, but the primary value of a review paper is to collect and properly describe publications related to a specific topic. And here, unfortunately, there is a big problem, because most of the literature references have incorrect numbers. In all cases that I have noticed, the correct reference number is 7 less than the number entered. So, for example, [21] reference number (page 2), regarding microbial communities from the American pika Ochotona princeps, actually refers to number [14] from Reference. So the whole work needs to be carefully checked in this regard and correct all wrong reference numbers.

I also don't understand why the order Blattodea is bolded in Table 1, in my opinion, it should be formatted the same as the other insect orders in this table.

Author Response

Thank you very much for your careful review and constructive comments. Our point-by-point replies to these comments are provided below. The line numbers refer to the revised manuscript, and all revisions are indicated in red.

Comments 1: This publication is quite interesting, but the primary value of a review paper is to collect and properly describe publications related to a specific topic. And here, unfortunately, there is a big problem, because most of the literature references have incorrect numbers. In all cases that I have noticed, the correct reference number is 7 less than the number entered. So, for example, [21] reference number (page 2), regarding microbial communities from the American pika Ochotona princeps, actually refers to number [14] from Reference. So the whole work needs to be carefully checked in this regard and correct all wrong reference numbers.

Response 1: Thank you very much for this comment! We are sorry for this mistake. The reference list is correct in the original submission. Unfortunately, it was changed after Pre-peer-review Check. We have modified the wrong reference numbers in the revised manuscript (lines 391–727).

Comments 2: I also don't understand why the order Blattodea is bolded in Table 1, in my opinion, it should be formatted the same as the other insect orders in this table.

Response 2: Thank you very much for this comment! This was also caused by the type setting of Pre-peer-review Check. We have revised the format of the order Blattodea in Table 1.

Thank you very much! Hopefully we have addressed all concerns.

All my best wishes,

Jing Chen

Key Laboratory of Zoological Systematics and Evolution

Reviewer 2 Report

Comments and Suggestions for Authors

The review entitled “Phylosymbiosis: the eco-evolutionary pattern of insect-symbiont interactions” demonstrates information on the phylosymbiosis of phylogenetically related species sharing similar microbial communities. Qin et al. present the different mechanisms that underlie phylosymbiosis in insects, including stochastic effects, evolutionary processes and ecological filtering. The review is well-written and gathers all available information on phylosymbiosis. It can be of great help to readers that wish to use it as a staring point on this subject. I fully recommend it for publication.

Some minor comments:

-        I am missing the strong effect of the host that shapes the microbiota in insects. I believe that authors should elaborate more on this. Here is a paper they could include: https://onlinelibrary.wiley.com/doi/10.1111/mec.16285

Although the review is mainly focused on the bacterial and fungal communities of insects, it would be useful to mention also the role of viruses in insects in the eco-evolutionary patterns. That would give to readers a global view of phylosymbiosis. Here is a paper they could use to include this part at the “Future directions…” part. https://www.researchgate.net/publication/329756445_Finer-Scale_Phylosymbiosis_Insights_from_Insect_Viromes

Author Response

Thank you very much for your careful review and constructive comments. Our point-by-point replies to these comments are provided below. The line numbers refer to the revised manuscript, and all revisions are indicated in red.

Comments 1: I am missing the strong effect of the host that shapes the microbiota in insects. I believe that authors should elaborate more on this. Here is a paper they could include: https://onlinelibrary.wiley.com/doi/10.1111/mec.16285

Response 1: Thank you very much for this comment! According to your kind suggestion, we have added the information in the “Phylosymbiosis in Insects” as follows: “Insect microbial community structures have been found to be correlated with environmental habitat, diet, sex, life stage, and host insect identity and phylogeny [30–32]. Some studies highlighted the strongest impact of insect species on the associated microbial communities [30,33].” (lines 75–78).

Comments 2: Although the review is mainly focused on the bacterial and fungal communities of insects, it would be useful to mention also the role of viruses in insects in the eco-evolutionary patterns. That would give to readers a global view of phylosymbiosis. Here is a paper they could use to include this part at the “Future directions…”. https://www.researchgate.net/publication/329756445_Finer-Scale_Phylosymbiosis_Insights_from_Insect_Viromes

Response 2: Thank you very much for this advice! According to your kind suggestion, we have added viral phylosymbiosis in the “Future Directions for Research on Phylosymbiosis in Insects” as follows: “In addition to bacterial and fungal communities, phylosymbiosis studies at the insect–virome level [170,171] will contribute to developing a comprehensive landscape of host–microbe symbioses.” (lines 362–365).

Thank you very much! Hopefully we have addressed all concerns.

All my best wishes,

Jing Chen

Key Laboratory of Zoological Systematics and Evolution

Reviewer 3 Report

Comments and Suggestions for Authors

see attachment

Author Response

Thank you very much for your careful review and constructive comments. Our point-by-point replies to these comments are provided below. The line numbers refer to the revised manuscript, and all revisions are indicated in red.

Comments 1: As a general comment: In sentences containing “that” or “that“ plus a verb, the first part can be deleted, beginning the sentence with “In” or “According to” or using the following verb as nomen. I suggest correction possibilities in lines 88, 91, 112, 154, 177, 195, 227, 249, 268, 300, 315 and 326. This shortens the draft and avoids a repetition of identical verbs such as show, demonstrate, suggest. This also avoids a citation of author names in the text.

Response 1: Thank you very much for this advice! According to your kind suggestion, we have made some modifications in the corresponding sentences. Please see “Phylosymbiosis in Insects” and “Mechanisms underlying Phylosymbiosis” in the revised manuscript (lines 88–92, 93–96, 107–109, 115–118, 153/154, 184–186, 234–236, 255–257, 314–316, 322–324, 333–334, 351–353).

Comments 2: Table 1 is confusing: Columns with text should be arranged in a left-hand fixation, all beginning in the first respective line. Columns with numbers should be arranged in a right-hand fixation with a space to the right-hand border. Thereby, the importance of the respective figure can be recognized easily. If the authors reduce the range of the column References, a two-line printing of one word is avoided. Blattodea should not be printed in bold.

Response 2: Thank you very much for this comment! We are very sorry for this confusion caused by the type setting of Pre-peer-review Check. According to your kind suggestion, we have revised the format of Table 1 in the revised manuscript.

Comments 3: 295: The authors should replace “gram” by “Gram”.

Response 3: Done.

Comments 4: 305-307: The authors should delete this sentence.

Response 4: Done.

Comments 5: 307: The authors should replace “Gilliamella” by “the gut symbiont Gilliamella in honey bees”.

Response 5: Done.

Comments 6: 542-805: The references contain too major types of mistakes.

  1. The authors should write English titles in lower case (lines 413, 508/509 and 517).
  2. The authors should write genus and species names in italics (lines 423, 604, 604/605, 607,620, 630/631, 639 and 640).

Response 6: Thank you very much for pointing out these mistakes! We are very sorry for them.

  1. We have checked allreferencescarefully and revised the English titles to lower case (lines 435, 561/562, 571/572). 
  2. We have revised genus and species names asitalics (lines 485, 648/649, 650/651, 664/665, 674/675, 683/684).

Comments 7: 632/633: The authors should abbreviate the journal.

Response 7: Thank you very much for this comment! We have checked the abbreviations of journal carefully. The abbreviations of “Memórias do Instituto Oswaldo Cruz” was revised as “Mem. Inst. Oswaldo Cruz” (lines 676/677). And “Tissue Cell” is actually the abbreviation of journal “Tissue and Cell” (lines 674/675).

Comments 8: Citation no. 134 doesn´t highlight the importance of the host immune system … It can be replaced by a more recent review: Guarneri, A.A.; Schaub, G.A.: Interaction of triatomines, trypanosomes and microbiota. In: Guarneri, A.A.; Lorenzo, M.G. (eds.) Triatominae – the biology of Chagas disease vectors. Springer Nature, New York, 345-386, 2021

Response 8: Thank you very much for this comment! According to your kind suggestion, we have revised the citation as Guarneri and Schaub (2021). In addition, we have chronologically sorted the references and exchanged the order of citation no. 140 and no. 141 (lines 662–665).

Thank you very much! Hopefully we have addressed all concerns.

All my best wishes,

Jing Chen

Key Laboratory of Zoological Systematics and Evolution
